# Influence of the Pre-Existing Defects on the Strain Distribution in Concrete Compression Stress Field by the AE and DICM Techniques

Nadezhda Morozova [1,*], Kazuma Shibano [1], Yuma Shimamoto [2] and Tetsuya Suzuki [3]

1 Graduate School of Science and Technology, Niigata University, Niigata 950-2181, Japan; f22e015j@mail.cc.niigata-u.ac.jp
2 Institute of Agriculture, Tokyo University of Agriculture and Technology, Tokyo 134-0006, Japan; simamoto@go.tuat.ac.jp
3 Faculty of Agriculture, Niigata University, Niigata 950-2181, Japan; suzuki@agr.niigata-u.ac.jp
* Correspondence: sakura131614@gmail.com

**Abstract:** This research investigates the influence of the pre-existing defects within concrete taken from the in-service irrigation structure on the strain distribution. The X-ray Computed Tomography (CT) technique is employed to investigate the internal concrete matrix and evaluate the defect distribution in it. The cracking system in a concrete matrix is detected as a damage type caused by the severe environment, and it is varied by the different degrees in all samples. The geometric properties of defects and their spatial location are obtained by image processing of CT images. The compression test with Acoustic Emission (AE) and Digital Image Correlation (DIC) measurements is conducted to analyze the fracture processes and acquire the damage spatial information. The AE signal descriptors are effective parameters for real-time detection and potential local damage monitoring. Moreover, the analysis of the DICM strain and displacement fields reveals the most potential fracture zones. The AE source location analysis indicated a connection between pre-existing defects and strain localization. The AE events and strain are high in the defect areas. Additionally, the amplitude and frequency of the AE events correlated with the location of the defects indicating that the structure weakness at that point leads to concentrated deformation development.

**Keywords:** concrete; in-service structure; uniaxial compression test; pre-existing defects; Acoustic Emission (AE); Digital Image Correlation Method (DICM); X-ray Computed Tomography (X-ray CT)

## 1. Introduction

Structure health monitoring is an important part of the optimal operation period maintenance of the civil concrete facilities. During the service period, structures are subjected to various loading and environmental conditions leading to the deterioration processes within the material. The primary danger to structure durability is posed by internal defects which can lead to structural unserviceability regarding safety and stability. Moreover, in the case of an earthquake or flood, the caused sudden loads can severely damage in-service structures and can lead to their final collapse. Concrete structures are popular worldwide structure types owing to their material versatility and relatively low cost: only since the 1950s, a large number of concrete structures have been made in Japan [1]. Concrete is a complex heterogeneous construction material with multiple discontinuities in the internal structure that accumulate under external forces and strongly affect the performance of the engineering structure. There are preliminary microcracks even prior to the loading caused by the differences in mechanical properties of coarse aggregate and cement paste and randomly distributed within the concrete matrix on the interface of both these phases [2]. Pre-existing defects developed in a structure during external factors' actions are the predominant origin of destruction initiation processes in concrete and define

their general deformational behavior. New defects may develop and coalesce with primary flaws leading to material deterioration and future structure failure [3]. The analysis of these processes is complicated, and the resulting damage degree cannot be evaluated by only mechanical properties' degradation. Therefore, there is a necessity to obtain more comprehensive information about the internal structure changes caused by defects such as cracks and voids developed during continuous loading to evaluate the damage degree precisely and establish efficient maintenance of concrete structures subsequently.

In this research, the analysis of the fracture behavior of concrete is conducted by such advanced non-destructive testing (NDT) techniques as Acoustic Emission (AE) and Digital Image Correlation Method (DICM) which allow recording micro-crack events and strain evolution in concrete. AE is a widely used elastic wave-based NDT method in structure monitoring to identify and characterize the damage process of concrete material [4]. By analyzing the AE signal descriptors recorded during fracture, the characterization of the nucleation and growth of the cracking pattern [5,6] and measurement of the deterioration of concrete structure [7,8] can be conducted. Wang et al. also used a combination of AE and CT techniques to study the fatigue fracturing evolution of granite with pre-existing flaws caused by repeated freezing-thawing cycles [9]. DICM is a full-field technique that can be applied to any type of material to investigate fracture evolution processes. In the work of [10], pre-peak and post-peak strain characteristics were described for rock specimens under uniaxial compression, and the observation that a major failure plane, due to strain localization, becomes noticeable only long after the peak stress occurs was made. Jerabeka et al. investigated the limits of accuracy of the DICM strain measurement system on polymeric materials under different environmental conditions and concluded the proper strain determination in terms of longitudinal and transverse strains as well as in terms of global average and local strains [11]. The DICM technique is also used to investigate the strain distribution fields and cracking phenomena in asphalt materials [12]. The study of fracture evolution in concrete has also been investigated in numerous research. Caduff and Mier analyzed the fracture behavior of three different types of concrete under compression [13]. Maruyama and Sasano investigated the aggregate–matrix interactions during drying in combination with the DICM and a fluorescent epoxy impregnation method (FEIM) and suggested that DICM can be applied to the damage evaluation of a concrete cross-section [14]. Because the flaws presented in the internal structure cause stress and strain redistribution, predicting the stress intensity near the crack is important in fracture mechanisms analysis. In research, the DICM was employed to calculate the stress intensity factor (SIF) for a crack in aluminum panels [15]. The advances of the DICM application to calculate the crack's SIF in concrete are discussed by Roux [16]. The fracture behavior of rock specimens with artificial flaws by DICM was observed by Zhou, and the differences between the failure mode of brittle and ductile material were mentioned [17]. The impact of a pre-existing flaw on rock damage behavior has been analyzed through laboratory-scale experiments in Shirole's research, and it was noticed that damage progression initiates at the tip of pre-existing flaw before propagating to other areas of the specimen [18]. The influence of frost damage on concrete mechanical performance was analyzed by Zhao [19]. By means of the DICM and AE techniques combination, Tong et al. revealed the evolution laws of generalized strain and AE events' location during crack initiation and propagation, and Li et al. analyzed the fractures on the macro and meso levels and instability behaviors of the jointed surrounding rock of a cavity under compression [20,21]. Zhu has investigated the influence of frost damage on concrete fracture based on the DICM and AE results and showed that the higher degree of frost damage is, the higher the crack initiation time and the propagation rate of the main fracture, resulting in a decrease in the ratio of high AE energy sources [22].

## 2. Materials and Methods

Based on DICM and AE technology, the fracture processes in concrete under uniaxial loading were investigated in this study. Damaged concrete was taken from the in-service irrigation structure affected by coupling severe environmental conditions, the defect of which was quantified by the X-ray CT method. The fundamental physical properties of the specimens were evaluated by measuring P-wave velocities and evaluating the dynamic modulus of elasticity using the resonance vibration method. Based on this result, only five samples have been selected for future investigation of pre-existing defect influence on strain distribution. The correlation between pre-existing defects and strain localization was revealed. The AE event allocation within the concrete matrix acquired by the AE source location analysis showed that the area near defects had a great concentration value. Moreover, the correspondence of amplitude and frequency of AE events to the defect location can evidence the magnitude and intensity of fracture in these points. The experimental flowchart is illustrated in Figure 1.

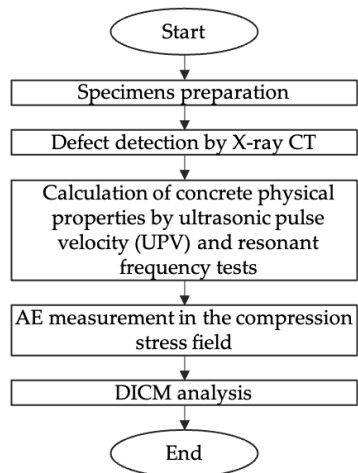

**Figure 1.** Experiment flowchart.

### 2.1. Specimen Preparation

The analysis of the influence of pre-existing flaws on the strain distribution in concrete was conducted on the samples drilled out from the in-service concrete irrigation structure located in Niigata, Japan, and operated for about 50 years. Shinkawa drainage pump station is the main agricultural irrigation facility in the district, but due to its deterioration over time, cracks have occurred all over the structure. In addition, Japan is a significantly subjected region to earthquake activity; therefore, there is a risk of causing extensive damage to this concrete structure [23]. According to the structure investigation reports [24,25], it was clear that the accumulated structural damage has a combination of Alkali–silica reaction (ASR) due to the use of reactive aggregate at the time of construction and salt damage due to the supply of chloride ions from the surface because the structure is located near seawater. In this research, the sampling size is twelve concrete core samples of cylindrical shape; the tested samples' dimensions are 150 to 204 mm in height and about 100 mm in diameter.

It is confirmed that the structure is damaged by the coupling effect of ASR and salt and frost deterioration mechanisms due to the long operation period and severe environmental conditions. Consequently, all samples tested have an initial damage degree in the form of internal discontinuities such as cracks and voids. Moreover, under visual investigation of samples, multiple flaws were detected on the surface after the coring test. It could be noted that, in this research, the term "pre-existing defect" is used to designate cracks or voids developed in a concrete structure during external factors affection such as environment and operation time. Figure 2 depicts the system of pre-existing defects detected visually.

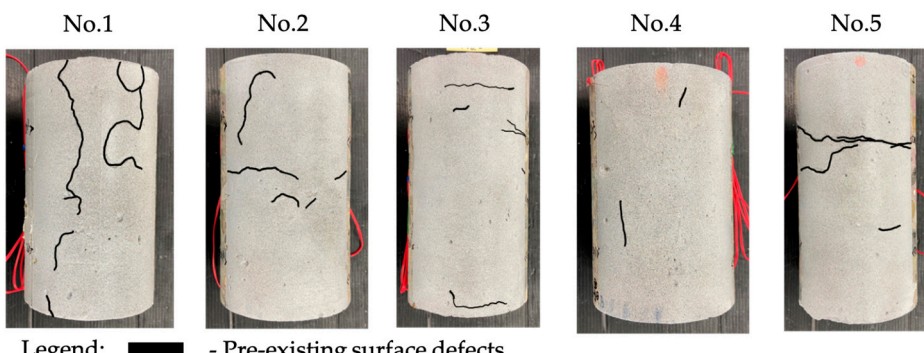

No.1　　　No.2　　　No.3　　　No.4　　　No.5

Legend: ▬ - Pre-existing surface defects

**Figure 2.** Pre-existing defect system.

## 2.2. Defect Detection by X-ray CT Method

In this study, the internal structure of the concrete samples was investigated by X-ray Computed Tomography (CT). This non-destructive technique allows the two- and three-dimensional (2D and 3D) visualization of the internal material media by obtaining individual projections, recorded from different viewing directions. In the results, a series of grayscale CT images are obtained, including information about the variation of attenuation coefficient within the scanning volume [26]. The X-ray CT images of the investigated concrete core's inner structure were received by the helical CT scans Aquilion ONE (TSX-301C/6A) (manufactured by TOSHIBA, Tokyo, Japan). The input conditions of X-ray CT scanning tests are presented in Table 1.

**Table 1.** X-ray CT test setting.

| Helical Pitch | Slice Thickness | Speed | Exposure | Recon Matrix | Field of View |
|---|---|---|---|---|---|
| | [mm] | [mm/rot] | [kV, mA] | [-] | [mm] |
| 51.0 | 0.5 | 0.5 | 120, 300 | 512 × 512 | 100–200 |

The extraction of concrete components including aggregates, voids and cracks is conducted from CT images of the central longitudinal sample surface by the binarization techniques. The Otsu and Max Entropy threshold calculation methods are employed to segment the concrete particles and increase the accuracy of this process [27,28]. Additionally, some adjustments are introduced in this research such as the extraction of only coarse aggregates and the distinguishing between cracks and voids by the circularity parameter. Figure 3 illustrates CT and segmented images.

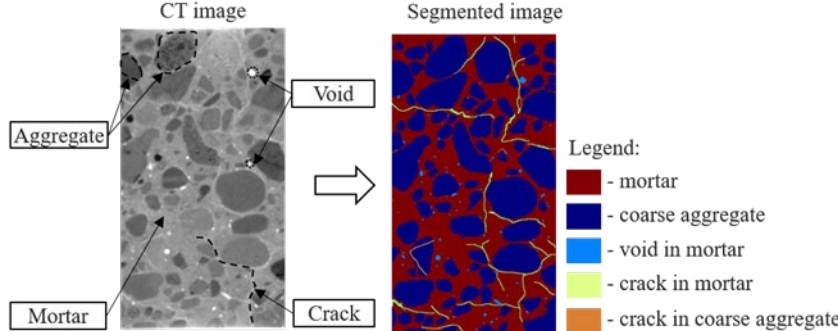

**Figure 3.** Concrete particle extraction.

A more precise explanation of this procedure can be obtained from our previous research on the visualization and quantification of concrete damage [29].

### 2.3. Physical Properties of Testing Core Samples by P-Wave Velocity and Resonance Vibration Methods

The elastic modulus of a material is an essential physical property that characterizes the deformation behavior of the material. This parameter can be determined by static or dynamic techniques. The first group of methods includes the direct loading of material and the acquisition of stress–strain responses where static modulus can be computed from the slope of the stress–strain curve in the elastic deformation range. In contrast, the dynamic modulus of elasticity can be received non-destructively which gives advantages in faster execution and without disruption of the material structural integrity. In this research, the physical properties of concrete core samples were investigated by the NDT methods. The ultrasonic pulse velocity (UPV) and resonant frequency tests were employed to investigate the internal concrete structure and calculate the dynamic modulus of elasticity. Both these techniques are most commonly used in the determination of materials' physical properties [30]. According to ASTM C 597-02, P-wave velocity was measured by the through-transmission technique using a UT device (Pundit Lab system manufactured by PROCEQ, Zurich, Switzerland) [31]. The input wave voltage and frequency were set to 50 V and 150 kHz, respectively; the receiver was set to double. Polyether polyol was used as a coupling agent between the transducers and the specimen's surfaces.

In accordance with JIS A 1127:2001, the fundamental longitudinal resonant frequency of the concrete was measured by the generation of vibration by the sweep excitation shaker at one end of the specimen and monitoring its reaction at the other [32]. The input voltage and sweeping time were set at 100 mV and 30 s, respectively. The sweep vibration condition is 500–20,000 Hz. The dynamic modulus of elasticity $E_D$ calculated from resonant frequency can be calculated by following Equation (1) [33]:

$$E_D = 4.00 \cdot 10^{-3} \frac{l}{A} m f^2, \tag{1}$$

where $f$ is the resonant frequency and $m$, $l$ and A are the mass, length and area of the testing sample, respectively.

### 2.4. AE Measurement in the Compression Stress Field

Under the load, energy stored in cracks is released as elastic waves which can be captured and monitored by AE piezoelectric sensors. The collected AE characteristics provide exact details on the processes by which damage develops inside the material. In this study, concrete specimens' degree of damage is examined using AE energy and its accumulation under loading. A configuration for performing a compression test with AE measurement is shown in Figure 4.

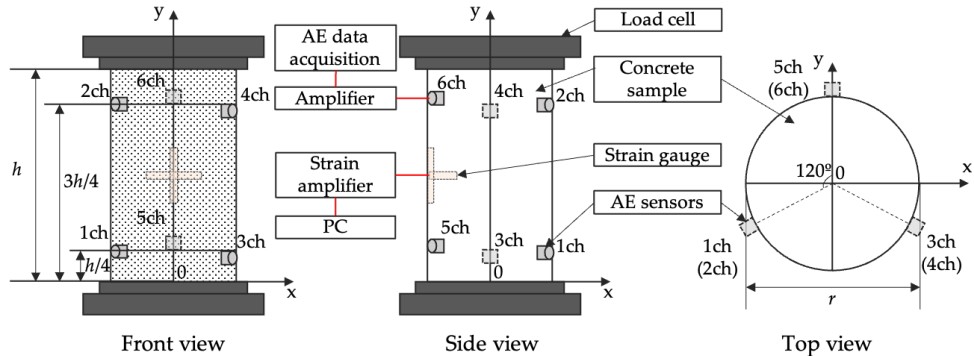

**Figure 4.** Compression test setup with AE measurement system.

The use of six piezoelectric AE sensors allowed the detection of AE activity. The AE sensor was an R15 with a 150 kHz resonance frequency. Two strain gauges installed on the specimen surface were used to gather strain data in both axial and lateral directions. An

AE measuring system (SAMOS made by PAC, New Jersey, US) was employed to extract the AE characteristics. All AE parameters were synchronized with the load values obtained from the compression testing machine. Table 2 demonstrates the AE test settings.

**Table 2.** AE test settings.

| Threshold | Sampling Rate | Main Amplifier | Pre-Amplifier | PDT, HDT, HLT | Pre-Trigger | Analog Filter |
|---|---|---|---|---|---|---|
| [dB] | [MHz] | [dB] | [dB] | [μs] | [μs] | [kHz] |
| 42 | 1 | 20 | 40 | 200, 800, 1000 | 256 | 5–400 |

Because the storage, dissipation, transformation and release of energy accompany the fracture processes in concrete under load, a series of indices have been proposed to assess concrete based on energy evolution. Strain energy is defined as the energy stored in a body due to applied load and can be calculated as the area under the stress–strain curve. This stored strain energy is transformed into the surface energy of the new crack faces, leading to microcracks development in the material's internal structure.

According to rate process theory [34], the AE energy release characteristics are evaluated by the cumulative AE energy occurrence frequency ratio [34] (Equation (2)),

$$f_e(U)dU = \frac{dE_{AE}}{E_{AE}}, \tag{2}$$

where $f_e(U)$ is the AE energy at the strain energy level $U$ (%), $dE_{AE}$ is AE energy per unit strain energy level $U$ and $E_{AE}$ is the total number of AE energy. The increment of strain energy level $dU$ is set to 0.5 N·m.

The high AE energy release events at the beginning loading stage can reflect the damage degree in concrete samples; thus, the initial AE energy release rate $\gamma$ is used to evaluate the damage of samples [35]. In this research, the analysis of the initial AE energy release rate is conducted at strain level in the range from 0 to $200 \times 10^{-6}$ (Equation (3)):

$$\gamma = \frac{E_{AE0\sim200}}{E_{AE}} \times 100, \tag{3}$$

where $E_{AE0\sim200}$ is cumulative AE energy in the strain levels $0$–$200 \times 10^{-6}$ and $E_{AE}$ is total AE energy in a series of the compression loading process.

To find the correlation between pre-existing defects and AE events location, the AE source location analysis is conducted to determine AE events' position within the concrete sample internal structure [36,37]. The corresponding amplitude and frequency of each AE event are also calculated.

### 2.5. DICM Analysis

In this research, the strain distribution on the specimen surface was determined by DICM. It provides a non-destructive and non-contact method for measuring the object surface deformation under load. With this technique, it is possible to obtain a complete reconstruction of the defect geometry which is important for the analysis of cracking processes in concrete. DICM is based on digital image processing and numerical computation. Figure 5a demonstrates the subset-based DICM technique where a reference square subset with sufficient intensity variations is selected from the reference image taken from an un-deformed state. After using set parameters and a certain optimization algorithm, the image in a deformed state is searched for whose intensity pattern is of maximum similarity with the reference subset [38,39].

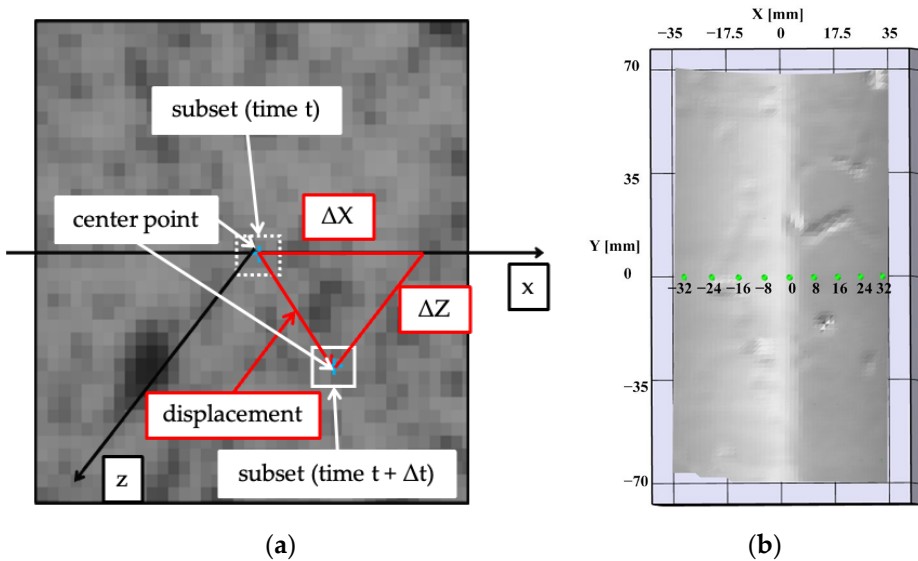

**Figure 5.** DICM: (**a**) DICM principle; (**b**) Strain measurement points.

Before the DICM test was conducted, the specimen surface was prepared by spraying black and white pigment to generate a random speckle pattern and provide strain and displacement measurements. In DIC, using an optimal speckle pattern is an important factor in reducing measurement noise and improving overall results of damage evolution in the material [40]. Therefore, the speckle pattern for concrete specimens was prepared in accordance with the guidelines developed by Correlation Solution, Inc [41]: the size of speckles ranged from $2 \times 2$ to $15 \times 15$ pixels, and the optimal speckle pattern was controlled by the exposure (electronic shutter of charge-coupled device (CCD) cameras). The DIC system consisted of two CCD cameras (Grasshopper3 GS3-U3-60S6M manufactured by Point Grey Research, Richmond, BC, Canada) with a spatial resolution of $2736 \times 2192$ pixels and two fixed-focus lenses (28 mm, Xenoplan manufactured by Schneider-Kreuznach, Dresden, Germany). The system was calibrated by a calibration board with a circular grid (size $12 \times 9$ mm, spacing 12 mm). During the calibration procedure, 21 pairs of images were recorded to ensure the accuracy of the calibration. For DICM analysis, a commercial software Vic-3D designed by Correlation Solution was used allowing accurate full-field 2D and 3D measurement of shape, displacement and strain of concrete specimens. Because DICM accuracy depends on numerous factors [42], the optimal conditions were set in accordance with the samples tested. User-defined inputs for image processing with DIC are illustrated in Table 3. The subset is the small group of pixels that allow us to track the changes by comparison of the reference image with later images of the deformed body taken at intervals of time. The subset size is set according to the sample dimension to reduce the noise level and loss of the spatial resolution of strain and displacement maps (sigma $\sigma$). The step size is pixel-spacing between centers of the subset, and it was set as $^1/_4$ of the subset size to get independent and non-repetitive data in our case. The filter size is set based on the optimal resolution of the data acquired.

**Table 3.** DIC test settings.

| Subset Size | Step Size | Filter Size | Correlation Accuracy, $\sigma$ |
|:---:|:---:|:---:|:---:|
| [pixel] | [pixel] | [pixel] | [-] |
| 37–47 | 10 | 29 | less 0.02 |

To evaluate the damage development, the strain and displacement metrics are used allowing the quantification of accumulated damage apart from visual strain distribution maps which provide only visual representations of strain fields. During the compression test, displacement fields ($U$, $V$ and $W$) at three directions ($X$, $Y$ and $Z$), the coordinates of each point of the speckled surface measured in mm and/or pixels, radial displacement ($dR$) and the sigma error of the correlation ($\sigma$) are derived. Moreover, the displacement fields are differentiated to calculate the normal and shear strains ($e_{xx}$, $e_{yy}$ and $e_{xy}$) and the principal strains ($e_1$ and $e_2$). In our research, only circumferential strain ($e_{xx}$) and radial displacement ($dR$) results of the DICM analysis are presented.

To receive behavior of surface strains and displacements during an entire loading and compare it to AE results, a time-series analysis of DIC data was conducted. Time-series analysis can show how variables change over time and how the data adjusts over the course of the data points as well as the final results and set order of dependencies between the data. The images collected in the test were processed using Vic-3D 8 software to calculate the strain development process in the concrete sample with time. For time-series analysis, nine points were selected on the horizontal line in the central part of the cylindrical sample. Calculation points were distributed at equal intervals of 8 mm. It is assumed that the deformation in this part reflects the whole fracture behavior of the concrete sample. The scheme is introduced in Figure 5b. Because DIC data are representative of the deformation process, the time-series data are considered on the strain energy level as well as the AE energy release rate allowing us to see the dependence of both parameters.

## 3. Results and Discussion

### 3.1. Defect System Characterization

The internal structure of concrete samples is depicted in Figure 6. The extracted concrete components such as coarse aggregates, voids and cracks can be evaluated tentatively by comparison of the density and orientation of internal structure particles.

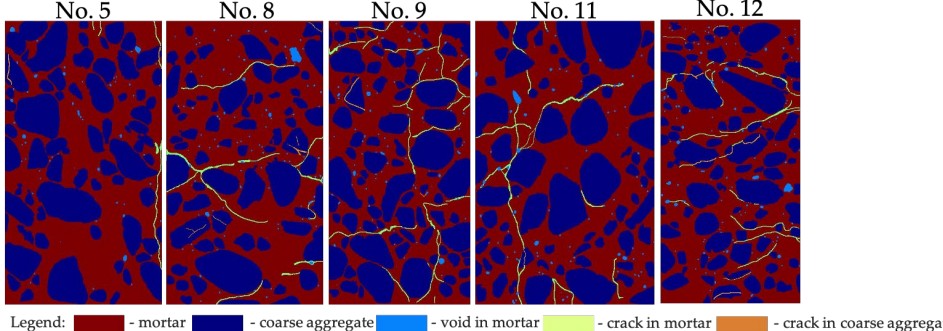

**Figure 6.** Internal defect system of compression test samples.

In accordance with color maps, each sample represents a heterogeneous structure with a unique allocation and orientation of coarse aggregates and voids within the concrete matrix. Samples No. 5 and No. 8 illustrate uneven coarse aggregate distribution density with big-size particles domination and gaps between them, whereas sample No. 9 shows the reverse tendency. To precisely assess the internal concrete structure, the corresponding components' geometric properties were calculated (Table 4). The average area rate of coarse aggregate in the entire sample set is 44.15% with average area and circularity of 111.10 mm$^2$ and 0.72, respectively. Void content consists of 0.78% on average, and the average area of the particle is 1.53 mm$^2$ with a circularity parameter of 0.97.

**Table 4.** Geometric properties.

| Sample Group | | Coarse Aggregate | | | Void in Mortar | | | Crack in Mortar | |
|---|---|---|---|---|---|---|---|---|---|
| | | Area Rate | Average Area | Circularity | Area Rate | Average Area | Circularity | Area rate | Average Area |
| | | [%] | [mm$^2$] | [-] | [%] | [mm$^2$] | [-] | [%] | [mm$^2$] |
| (No. 1–No. 12) | Average | 44.15 | 111.10 | 0.72 | 0.78 | 1.53 | 0.97 | 1.55 | 17.94 |
| | Max | 50.89 | 153.39 | 0.76 | 1.09 | 2.05 | 0.98 | 2.42 | 29.67 |
| | Min | 35.23 | 91.04 | 0.68 | 0.44 | 1.21 | 0.96 | 0.31 | 7.35 |
| | SD | 4.43 | 18.86 | 0.02 | 0.19 | 0.24 | 0.00 | 0.57 | 5.39 |
| Selected | No. 5 | 50.89 | 137.27 | 0.71 | 0.44 | 1.24 | 0.98 | 0.31 | 7.35 |
| | No. 8 | 41.92 | 105.07 | 0.73 | 0.78 | 1.40 | 0.98 | 1.49 | 15.76 |
| | No. 9 | 46.28 | 103.25 | 0.74 | 0.79 | 1.60 | 0.97 | 2.42 | 14.73 |
| | No. 11 | 45.27 | 109.69 | 0.74 | 0.73 | 1.39 | 0.97 | 1.48 | 19.60 |
| | No. 12 | 40.01 | 94.90 | 0.70 | 0.95 | 1.59 | 0.97 | 1.58 | 11.51 |
| Selected | Average | 44.87 | 110.03 | 0.72 | 0.74 | 1.44 | 0.98 | 1.45 | 13.79 |
| | Max | 50.89 | 137.27 | 0.74 | 0.95 | 1.60 | 0.98 | 2.42 | 19.60 |
| | Min | 40.01 | 94.90 | 0.70 | 0.44 | 1.24 | 0.97 | 0.31 | 7.35 |
| | SD | 3.76 | 14.43 | 0.02 | 0.17 | 0.13 | 0.00 | 0.67 | 4.13 |

The pre-existing defect system exists in the form of mortar cracks in all samples tested. Regarding the entire sample set, the average area rate of cracks in mortar is 1.55%, and the average crack area is 17.94 mm$^2$. Sample No. 5 demonstrates a slight damage degree by only one continuous crack near the lateral surface (area rate is 0.31%). Other samples show a complicated damage system with randomly oriented cracks. Samples No. 9 and No. 12 can be characterized as concrete with a heavy damage degree with a crack area rate of 2.42% and 1.58%, respectively. Comparing Figures 1 and 5, there are some correlations between internal and surface defects regarding location. Therefore, by analyzing strain and displacement localization on the sample surface, the corresponding development of internal defects can be assumed.

*3.2. Concrete Physical Properties: Comparison of Accumulated Cracking Damage and the Dynamic Modulus of Elasticity*

The physical properties of the samples are shown in Table 5. According to the NDT results, all samples can be characterized as damaged with an average pulse velocity of 1951 m/s which is twice lower than the pulse velocity in good-quality concrete (4000 m/s) [43]. The result of resonant frequency tests also shows a low value of the elastic parameter: the average frequency and corresponding $E_D$ value are 7587 Hz and 15.9 GPa, respectively. The obtained results are depicted in Figure 7. This figure demonstrates the correlation between accumulated cracking damage and $E_D$ ($R^2$ = 0.51). All samples show various damage degrees under different physical properties: sample No. 5 has the highest $E_D$ (28.2 GPa) under low crack density (0.31%), whereas the internal structure of sample No. 9 demonstrates a high degree of cracking damage (2.42%) and poor value of dynamic modulus of elasticity ($E_D$ is 12.7 GPa), consequently. Both can be regarded as slightly and heavily damaged specimens. As for the rest of the samples, the dispersion of values is high, and they are placed in the middle part of the graph with a little shift to the left that demonstrates their damaged condition. Based on this result, we decided to conduct an evaluation of pre-existing defects' influence on the strain distribution for five following samples: No. 5, No. 8, No. 9, No. 11 and No. 12. As can be seen, samples No. 8, No. 11 and No. 12 have about the same value of accumulated cracking damage (1.49%, 1.48% and 1.58%, respectively) but a different value of dynamic modulus of elasticity $E_D$ (14.9 GPa, 19.8 GPa and 11.6 GPa, respectively). To clearly understand this phenomenon, the compression test with AE measurement and strain localization will be conducted for these selected samples.

**Table 5.** Physical properties.

| Sample Group | | Pulse Velocity, $V_p$ | Resonant Frequency | $E_D$ |
|---|---|---|---|---|
| | | [m/s] | [Hz] | [GPa] |
| (No. 1–No. 12) | Average | 1951 | 7587 | 15.9 |
| | Max | 3730 | 13,561 | 36.6 |
| | Min | 977 | 4553 | 5.0 |
| | SD | 730 | 2630 | 8.6 |
| Selected | No. 5 | 3730 | 11,529 | 28.2 |
| | No. 8 | 2035 | 7155 | 14.9 |
| | No. 9 | 1226 | 6234 | 12.7 |
| | No. 11 | 2650 | 9470 | 19.8 |
| | No. 12 | 977 | 5567 | 11.6 |
| Selected | Average | 2124 | 7991 | 17.4 |
| | Max | 3730 | 11,529 | 28.2 |
| | Min | 977 | 5567 | 11.6 |
| | SD | 999 | 2208 | 6.1 |

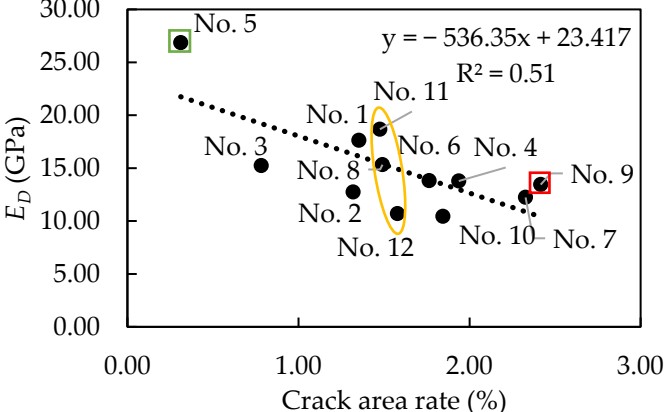

**Figure 7.** Relationship between accumulated cracking damage and $E_D$.

*3.3. Fracture Behavior*

The mechanical properties and AE measurement results of the testing concrete samples are represented in Tables 6 and 7, respectively.

**Table 6.** Mechanical properties.

| Sample Name | Compressive Strength, $\sigma$ | Maximum Strain, $\varepsilon$ ($\times 10^{-6}$) | Initial Tangent Modulus, $E_0$ | Secant Modulus, $E_c$ | Strain Energy, $U$ |
|---|---|---|---|---|---|
| | [N/mm²] | [-] | [GPa] | [GPa] | [J] |
| No. 5 | 18.3 | 1749 | 11.7 | 10.4 | 19.4 |
| No. 8 | 12.7 | 2309 | 2.8 | 5.5 | 14.4 |
| No. 9 | 12.0 | 1873 | 5.3 | 6.4 | 13.7 |
| No. 11 | 14.9 | 737 | 5.2 | 20.2 | 5.1 |
| No. 12 | 14.9 | 1797 | 5.2 | 8.3 | 20.1 |
| Average | 14.5 | 1693 | 6.0 | 10.2 | 14.5 |
| Max | 18.3 | 2309 | 11.7 | 20.2 | 20.1 |
| Min | 12.0 | 737 | 2.8 | 5.5 | 5.1 |
| SD | 2.2 | 518 | 3.0 | 5.3 | 5.4 |

**Table 7.** AE test results.

| Sample | Total AE Hits | Total AE Energy | Initial AE Energy Release Rate $\gamma$ (0–200) |
|---|---|---|---|
| | **[hit]** | **[V$^2$]** | **[%]** |
| No. 5 | 33,133 | 13,319.6 | 0.18 |
| No. 8 | 65,340 | 28,661.8 | 0.06 |
| No. 9 | 119,056 | 24,480.3 | 0.45 |
| No. 11 | 56,582 | 21,516.5 | 0.31 |
| No. 12 | 74,888 | 15,215.5 | 0.43 |
| Average | 69,800 | 20,638.8 | 0.29 |
| Max | 119,056 | 28,661.8 | 0.45 |
| Min | 33,133 | 13,319.6 | 0.06 |
| SD | 28,253 | 5707.5 | 0.15 |

All samples had an initial damage degree confirmed before the compression test was conducted. Thereby, low mechanical properties were expected. The compressive strength of concrete samples did not exceed 20 N/mm$^2$ and ranged from 12.0 N/mm$^2$ (sample No. 9) to 18.3 N/mm$^2$ (sample No. 5). The strain value also was observed in all samples: maximum and minimum values are in samples No. 8 (2309 $\times$ 10$^{-6}$) and No. 11 (737 $\times$ 10$^{-6}$), respectively. Strain energy is stored in the material during work of external load and is needed for the crack formation and propagation processes. Sample No. 12 has the highest strain energy value (20.1 J), whereas sample No. 11 illustrates the lowest value (5.1 J).

The number and distribution of pre-existing defects in each concrete sample are unique and vary in different degrees, and therefore, the AE hits and AE energy present obvious distinctions in obtained values. The AE hit numbers in samples No. 9 (119,056 hits) and No. 12 (74,888 hits) show a high degree of crack activity under load which corresponds to the high internal damage degree in these samples. Slightly crack-damaged sample No. 5 demonstrates low cracking activity with a total number of AE hits and AE energy of 33,133 hits and 13,319.6 V$^2$, respectively. The AE energy response and cumulative energy curve versus strain energy level during the loading process are shown in Figure 8.

As can be seen, the AE energy release generated by concrete deformation is represented as a process of constant energy accumulation and intermittent energy discharge. Each specimen illustrates the diverse AE energy release trend which relates primarily to the distribution of internal defects and their severity. Generally, the AE energy release in non-damaged concrete has the following stages: the sparse activity period is characterized by the absence of any significant AE signals and the closure of existing micro-cracks within the internal structure. After this stage, the slow growth period occurs of the AE activity where internal damage generation starts. With stress increase, the rapid growth of the AE energy and accumulative events can be observed as the result of the propagation, coalescence and penetration crack processes. At the stress peak, AE energy reaches its highest value and afterward gradually reduces due to the multiple crack penetration processes, and the final failure of the material occurs. Sample No. 5 (Figure 8a) shows a similar trend as described above: a gradual step-up trend with activity absence at the beginning stage and its increasing with the load increasing. The cumulative AE energy curve of other samples has a greater slope meaning in faster AE energy increase that reflects the more serious damage degree in the material. AE activity in samples No. 9 and No. 12 starts to generate earlier and shows an obvious high intensity at the beginning of loading compared to the rest of the samples (Figure 8c,e). Sample No. 11 (Figure 8d), similar to No. 5, has a silent stage at the initial loading, but in the middle, it has a great emission event corresponding to the main fracture occurrence. The initial AE energy release $\gamma$ in all samples varies insignificantly: $\gamma$ does not exceed 1% for all samples. Thus, this parameter cannot be used for damage

degree investigation in our case. Consequently, these concrete samples demand a more complex assessment of fracture behavior regarding strain and displacement evolution.

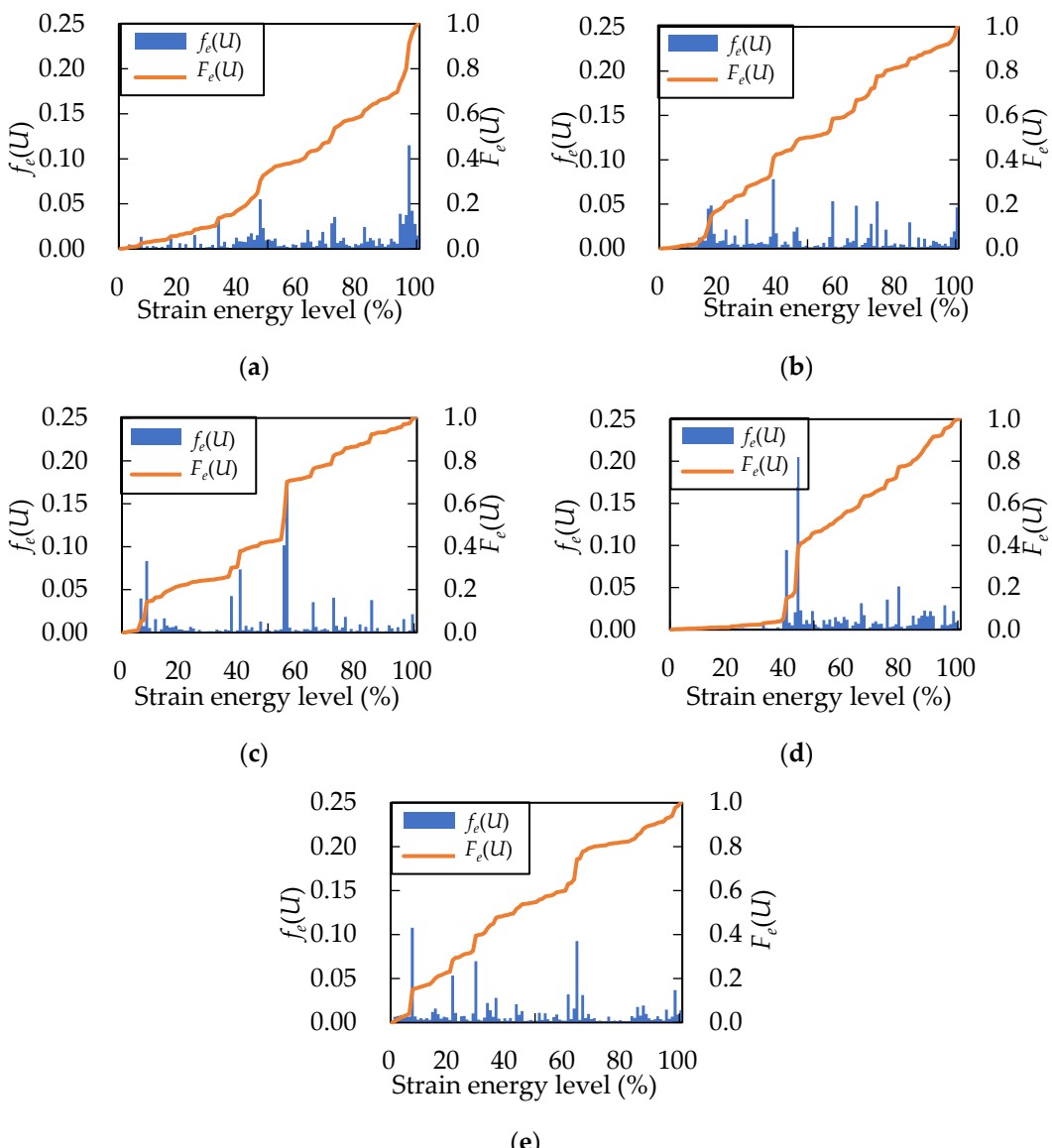

**Figure 8.** AE energy release rate $f_e(U)$ and cumulative AE energy $F_e(U)$: (**a**) AE energy release in sample No. 5; (**b**) AE energy release in sample No. 8; (**c**) AE energy release in sample No. 9; (**d**) AE energy release in sample No. 11; (**e**) AE energy release in sample No. 12.

### 3.4. Strain Localization

The DICM results in terms of strain fields in the circumferential direction $e_{xx}$ and radial displacement $dR$ are provided. Using the method, the strain of any point within the observation area of the concrete is computed, so the initiation and development of the cracks are investigated through the full-field contour images where the $e_{xx}$ and $dR$ change under different loadings. Figure 9 shows the strain and displacement fields. The color scale bar presents positive and negative values corresponding to different colors, and the direction of movement can be determined in accordance with a coordinate system depicted in Figure 5: a positive value indicates that the sample is subjected to expansion and the displacement is moving to the radial direction in the color map. In this figure, the strain and displacement contours are represented at different strain energy levels which are inflection points on the AE energy trend (Figure 8).

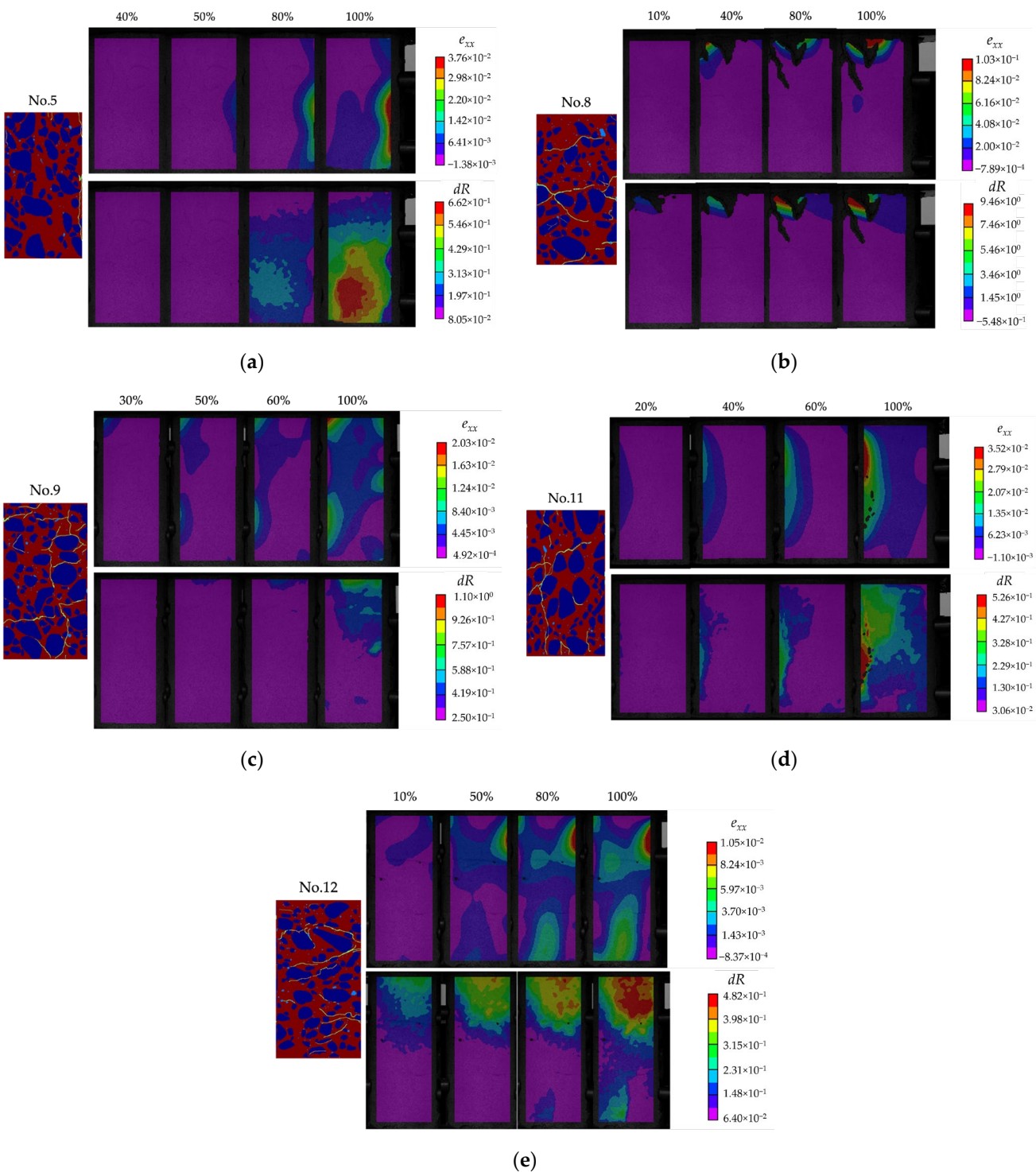

**Figure 9.** Circumferential strain $e_{xx}$ and radial displacement $dR$: (**a**) Distribution in sample No. 5; (**b**) Distribution in sample No. 8; (**c**) Distribution in sample No. 9; (**d**) Distribution in sample No. 11; (**e**) Distribution in sample No. 12.

For sample No. 5 (Figure 9a), the strain in the x and z directions appeared to be a localization phenomenon at 50% of the strain energy level and the strain localization area concentrated along the center of the pre-existing flaw in the internal structure during the loading process. It indicates the main crack pattern which started from this location. The AE energy release at the strain energy level of 50% showed a sudden increase which means the significant crack development and the final failure corresponds to the high AE energy

emission event near 100% of the strain energy level (Figure 8a). The localized displacement has also started from the 50% of strain energy level which is concentrated in the central part of the sample and moved to the bottom side implying the expansion of the radial surface caused by the pre-existing defects' modifications. The strain $e_{xx}$ and radial displacement $dR$ of damaged and non-damaged areas illustrate the significant difference in the values from the crack occurrence event (40% of the strain energy level). In sample No. 11, the same strain distribution can be mentioned: the circumferential strain localizes near the internal defects forming the main fracture pattern (Figure 9d). However, here, strain started to localize before the high AE event occurred. AE energy release had a sudden skip near 40% of strain energy level that demonstrates that fracture processes evolved faster (Figure 8d).

Sample No. 8 illustrates the different strain and displacement phenomena: the main failure evolved on the specimen's upside and the edge popping out is clearly visible (Figure 9b). This process started at 20% of the strain energy level and dominated until the final failure as the mainly strain-concentrated and deformable area. According to the AE energy release behavior, the fracture events accompanied the loading process which is expressed in the continuous AE emission (Figure 8b). Rapid AE energy release occurred at 20% of the strain energy level corresponding to the edge deformation beginning, and at 40%, the popping-out was completed. Afterward, this place became a prevalent weak zone where cracks developed.

In sample No. 9 (Figure 9c), strain localization can be detected immediately in different parts: the edge and back sides of the concrete core. It also can be mentioned that the cracking development processes occur faster and the coalescence can be detected, which says about an unstable failure stage leading to rapid failure. After a sudden AE energy release event corresponding to the crack development, the difference is changed significantly, which is about major deformation processes in the material. Sample No. 12 also has multiple strain localization points which develop near the weak areas caused by pre-existing defects (Figure 9e).

### 3.5. Signal Location Analysis

Applying the AE source location analysis, AE sources detected during the entire loading process were identified and the results are represented in Figure 10.

The background of each graph depicts the corresponding internal sample structure where pre-existing defects are clearly detected. The signal parameters were sorted by amplitude and peak frequency. In this research, the amplitude analysis was conducted for three bands: 42 to 59 dB, 60 to 69 dB and 70 to 99 dB. In peak frequency analysis, the signals were sorted out by 10 to 59 kHz, 60 to 99 kHz and 100 kHz and higher. The location of each AE event and its intensity are clearly observed. During the concrete deformation and fracture process, the AE frequency spectrum range was very wide. The distribution characteristics of frequency–amplitude greatly differed under various systems of pre-existing defects in the internal structure of the material. It is investigated that the amplitude value of low-frequency signals (10–59 kHz and 60–100 kHz) was bigger than that of high-frequency (bigger than 100 kHz). Usually, high-frequency waves are generated from small defects relating to the damage process beginning, whereas low-frequency ones can be produced only from large cracks developed during the final failure [44]. The dispersion of the signals in samples No. 5 and No. 8 is higher and distributed through sample height (Figure 10a,b). In sample No. 5 (Figure 10a), the concentration point is observed: AE signals of high amplitude (70–99 dB) are allocated near the strain concentration point and where a defect exists that is evidence of the crack initiation point. The high number of AE events of low frequency (10–59 kHz and 60–99 kHz) is also located at this point corresponding to the main crack initiation and propagation. In sample No. 8 (Figure 10b), the AE events of high magnitude and low frequency can be noticed in the upper side of the specimen caused by the popping out of the concrete piece. The main fracture occurred near the lower sample side where the AE event concentration was denser. In other samples, the AE event accumulation illustrates compact behavior with a higher degree of event quantity. There is

no obvious event localization point; events of different amplitude and frequency bands are presented in each part of the samples. The presented internal structure of samples also shows an uneven allocation of pre-existing defects that influence AE behavior. From Figure 10, the main deformation parts can be distinguished as follows: the majority of AE events in sample No. 9 are distributed in the central part of the cylinder (Figure 10c), the upper and left side of the specimen (Figure 10d) and on both sides of the central part (Figure 10e). The AE event distribution pattern in these samples replies to the internal defect system contour: events are located in the upper part of sample No. 11 and move to the lower one by the left side where a longitudinal crack exists (Figure 10d). In Figure 9e, two main concentration clouds of AE events can be observed which also correspond to the major crack allocation positions. It can be mentioned that these samples have some portion of low-frequency high-amplitude AE signals. It can be related to the fact that the coalescence of many cracks under relatively high stresses can involve cracks with diverse smaller scales. It can indicate the fracture instability stage of concrete [45].

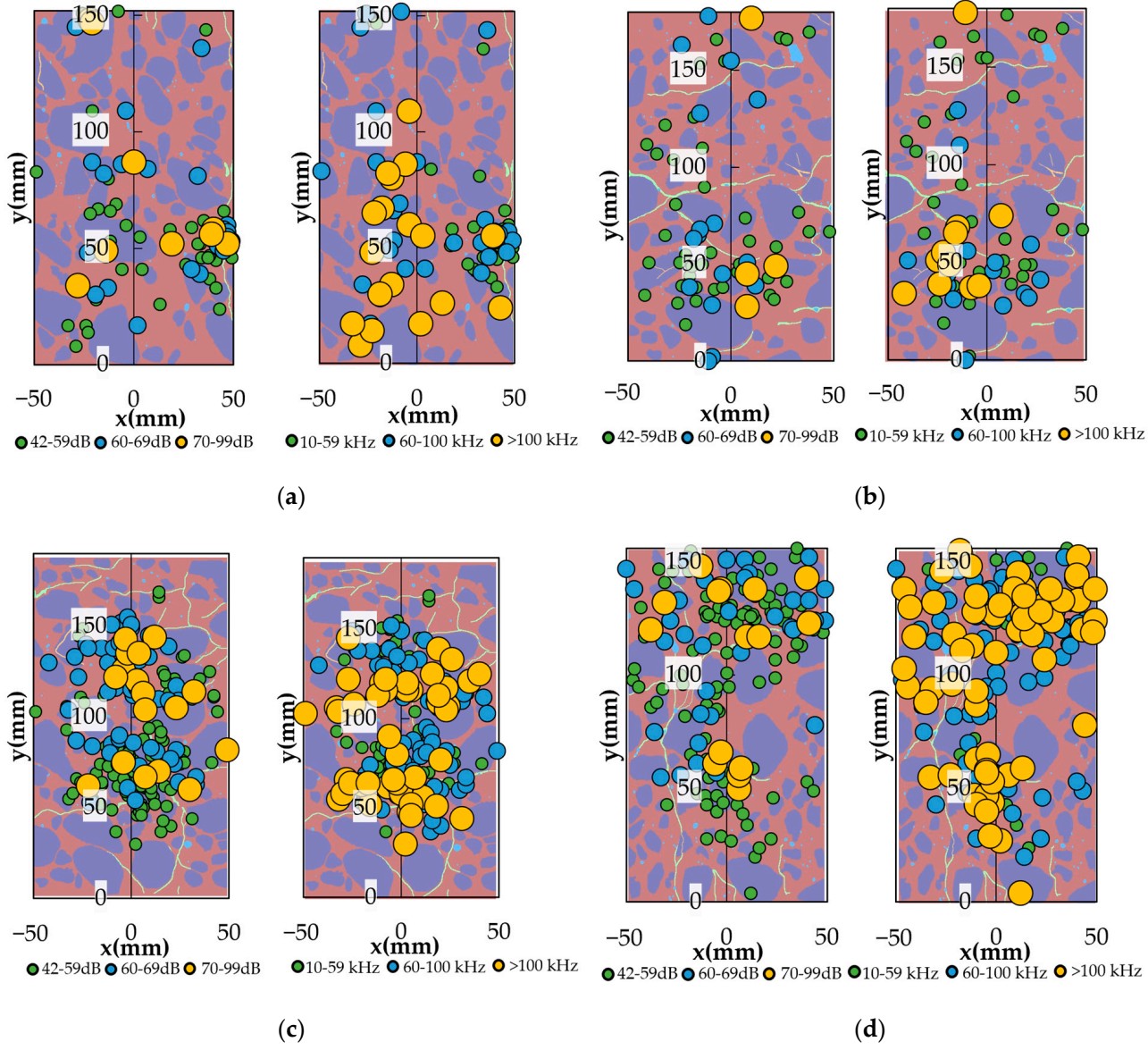

**Figure 10.** *Cont.*

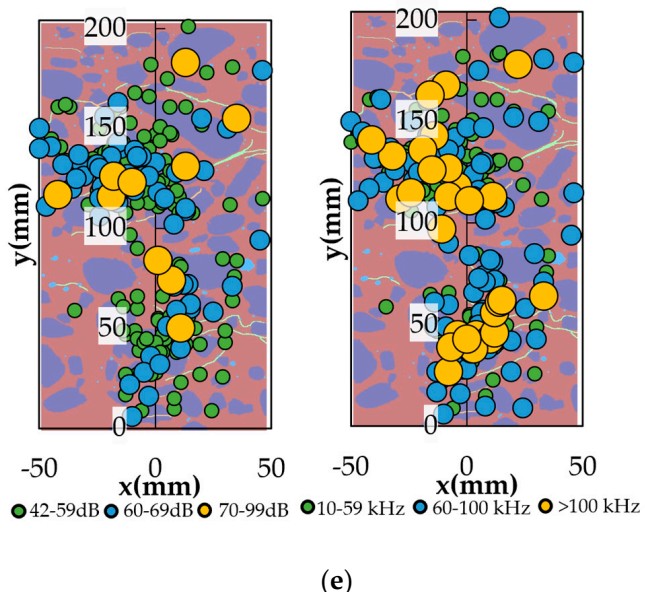

(**e**)

**Figure 10.** AE source location analysis: (**a**) Amplitude and Frequency of AE events in sample No. 5; (**b**) Amplitude and Frequency of AE events in sample No. 8; (**c**) Amplitude and Frequency of AE events in sample No. 9; (**d**) Amplitude and Frequency of AE events in sample No. 11; (**e**) Amplitude and Frequency of AE events in sample No. 12.

To get a more comprehensive understanding of the signal source locations, we decided to apply an image-gridding method for establishing a relationship between defects and AE event characteristics. The binary images of cracks extracted were divided into a grid of smaller cells, and relative crack density was counted in each grid cell. The grid scheme is presented in Figure 11. In this research, the number of cells in each sample was equal to nine, and their size was the same varying in each sample in accordance with its dimensions. The same procedure was performed for the computation of AE event location and characteristics in each cell. Figure 12 depicts the observed dependence. The vertical axis represents the different units corresponding to each value on the horizontal axis such as the number of AE events, amplitude in dB and peak frequency in kHz. The average value of crack density for samples No. 1 to No. 12 and for an investigated sample is also presented on each graph as red and black lines, respectively.

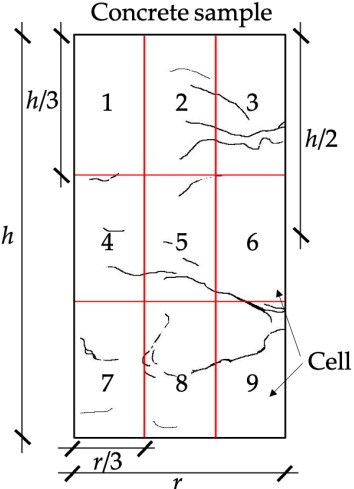

**Figure 11.** Image-gridding method.

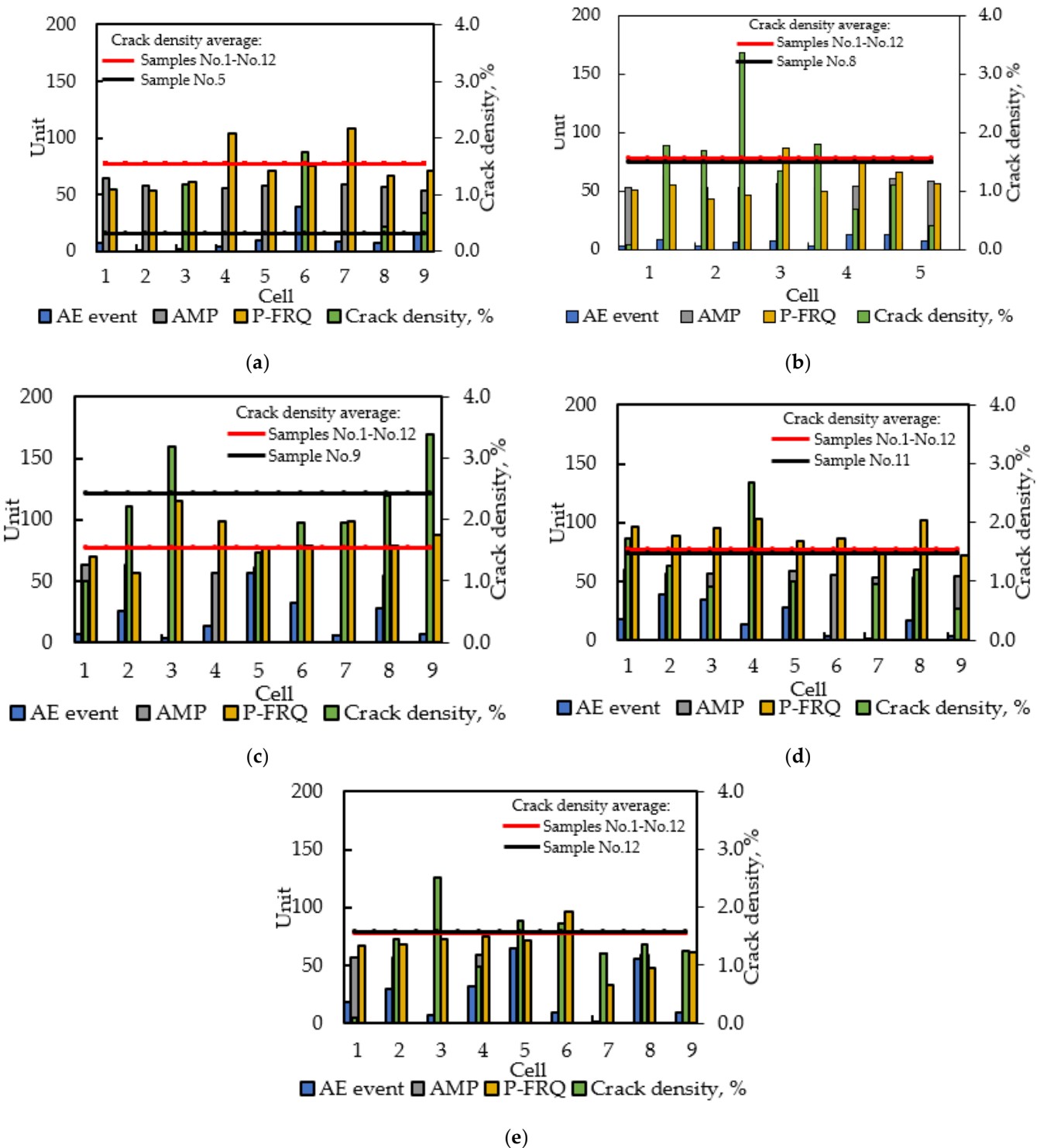

**Figure 12.** Counts of AE parameters and crack density in the grid: (**a**) sample No. 5; (**b**) sample No. 8; (**c**) sample No. 9; (**d**) sample No. 11; (**e**) sample No. 12.

As can be seen, sample No. 5 has low cracking damage concentrated only on the right side of the sample (Figure 12a). The respective cells show a high enough value of amplitude and peak frequency, especially in cell No. 6 where relative crack density is maximum. In contrast, other samples have considerable damage degrees distributed on the whole sample area. In sample No. 8, the amplitude is distributed at the same level of about 50 dB whereas the peak frequency ranges more significantly that demonstrate the different crack growth rates in each part of the sample. Pre-existing cracks in cell No. 4 contribute a higher

amount and their development is slower which is reflected by the low peak frequency AE signals content. In contrast, in adjacent cells of No. 5 and No. 7, the increase in AE events number, amplitude and peak frequency can indicate the crack development in these directions. The distribution tendency of AE events and their characteristic in samples No. 9 and No. 12 have similar features: a significant number of AE signals in central cell No. 5 of high amplitude (more than 50 dB) and peak frequency (about 80 kHz) correspond to active fracture processes occurrence in this area. Sides of the specimens in terms of adjacent cells No. 4 and No. 6 also show the considerable parameters of AE events that can be related to the main failure pattern as side expansion. It can be mentioned that the peak frequency in samples No. 9 and No. 11 demonstrate higher average values than in others which can be related to the crack's orientations. From Figure 5, the vertical cracks are prevalent in both these samples, whereas in samples No. 8 and No. 12, the horizontal cracks number is dominant. It was found that the crack propagation velocity parallel to the loading axis is faster than that of the perpendicular one to the loading axis [46]. Therefore, a high peak frequency can be expected in concrete samples with vertically oriented cracks.

### 3.6. Time-Series Analysis

The comprehensive concrete fracture behavior is described by the AE and DICM parameters from time-series data (Figure 13).

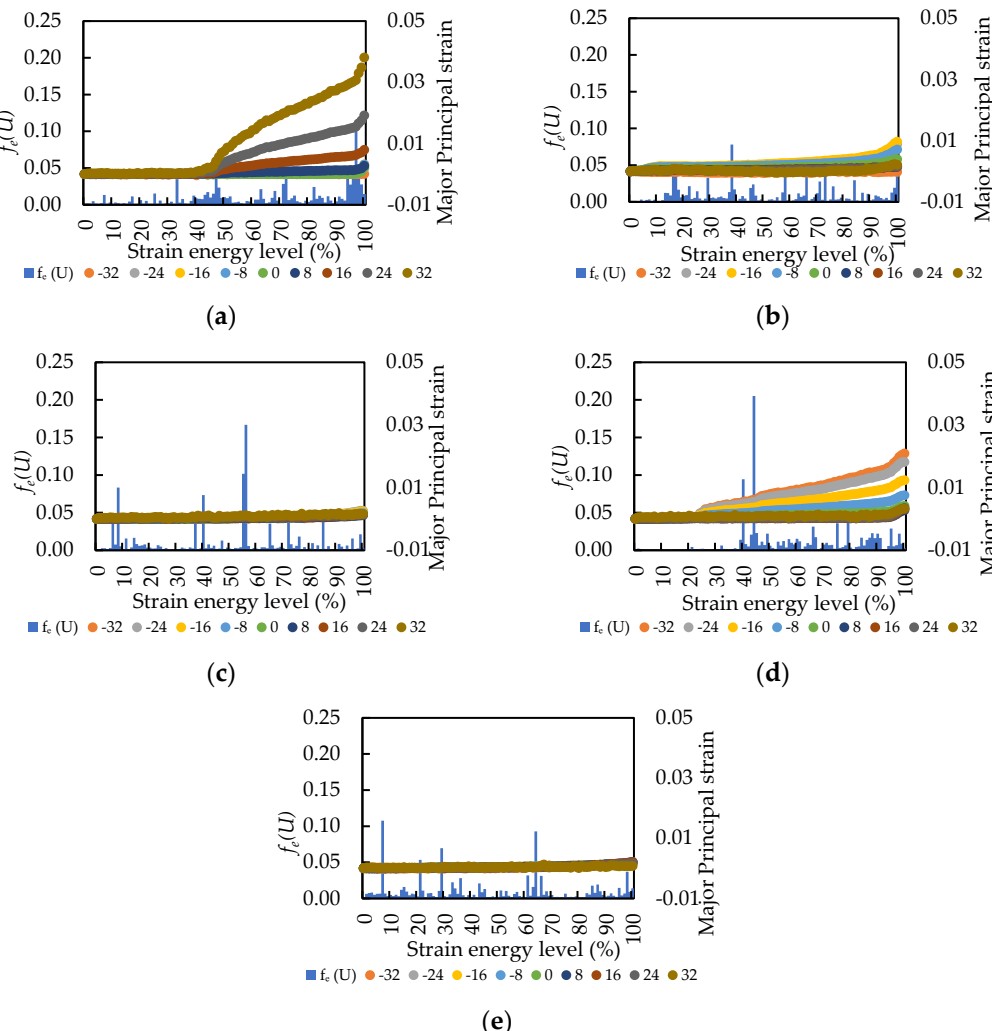

**Figure 13.** AE and DICM time-series data: (**a**) sample No. 5; (**b**) sample No. 8; (**c**) sample No. 9; (**d**) sample No. 11; (**e**) sample No. 12.

The vertical left axis of the graph corresponds to the major principal strain measured by DIC, and the vertical right axis is the AE energy release rate. The comparison of both parameters is shown on the strain energy level. The major principal strain was extracted in the nine locations on the horizontal line in the central sample's part during the entire compression loading (Figure 5b). As can be seen from Figure 13, each sample has a unique deformation trend depending on the specific pre-existing fracture distribution. Moreover, there is a relationship between AE energy release activity and deformation. As both AE and DICM are non-destructive techniques allowing the monitoring and visualization of the interior and exterior damage evolution in the material, the early fracture processes and their development can be detected by the high AE energy release and strain concentration in the area of the material. Therefore, a precise analysis of fracture behavior can be conducted based on these results. Using time-series analysis, the relationship between both parameters can be visible. Sample No 5 (Figure 13a) showed the sudden AE emission of high intensity at the strain energy level of 40%, where the strain concentration can be distinguished at the same level. The most contributed part is the right sample part, where the main fracture can be concluded. Sample No. 11 (Figure 13d) also illustrated the clear deformation process starting at 20% of the strain energy level. However, the main AE activity can be found at the 40% of strain energy level, which can be due to the defects distribution within the concrete matrix: small energy for crack development released that can be the evidence of severe conditions of pre-existing defects system. The slope change can be seen at the 40% strain energy level, which means the rate of fracture development is increasing. Both these samples demonstrate the obviously dominated zone of deformation, which is correlated with pre-existing defects. In addition, the sudden slope change can be seen at 100% of the strain energy level, which depicts the final stage of sample failure. Sample No. 8 demonstrates the earlier deformation behavior at the initial loading stage corresponding to the popping out of the coarse aggregate on the upside of the sample. The slope also changed after the high AE energy release (40% strain energy level) and continued until final failure with deformation domination on the right sample side. Samples No. 9 and No. 12 (Figure 13c,e) show the continuous strain concertation in the whole central part that is correlated with the pre-existing defect system distribution. The same slope-changing phenomena can be mentioned after a significant AE energy release event as in the previous samples.

According to time-series results, the correlation between AE and DICM parameters can be attributed to the intense AE emission and strain concentration. Both these aspects indicate the occurrence and localization of fracture activity, the presence of defects within the concrete matrix, the severity of pre-existing defect systems and the initiation and development of fractures. The changes in slope after significant AE energy release events suggest changes in fracture mechanisms, accelerated fracture development rates and the propagation of fractures leading to further deformation and eventual failure. The combination of AE and DICM data provides a comprehensive understanding of concrete fracture behavior and enables a precise analysis of fracture processes and their correlation with pre-existing defects.

## 4. Conclusions

The investigation of the influence of pre-existing defects on strain distribution in compression stress field is conducted. The internal concrete structure taken from the in-service irrigation structure was investigated by the X-ray CT method. Concrete damage was quantified by image analysis and the segmentation of concrete particles. The physical properties of concrete samples were investigated by such NDT methods as pulse velocity and the resonant frequency technique, and the dynamic modulus of elasticity was calculated. A compression test with measurement of AE behavior and deformation phenomena was carried out. Strain localization and a displacement contour map were obtained by the DICM technique. Moreover, the AE source location analysis was employed to acquire the AE event allocation. From the results, the relationship between concrete basic properties

$E_D$ and characteristics of AE and DICM in the compression field can be observed and the main conclusions of this study are summarized as follows:

1. The X-ray CT technique is suitable for the investigation of internal concrete structures and can be used to quantify the damage degree by calculating the geometric properties of concrete components. The prevalent damage type in concrete is detected in the form of mortar cracks in all samples case: crack density varies from 0.31 % (sample No. 5) to 2.42 % (sample No. 9).

2. The deteriorated condition of concrete samples is confirmed by the low physical property values calculated by NDT: average pulse velocity and resonant frequency values are 1951 m/s and 7587 Hz for all samples, respectively. Dynamic modulus of elasticity, $E_D$, calculated from the resonant frequency, shows a good correlation with accumulated concrete cracking damage: the stronger the cracking system, the lower modulus of elasticity. In slightly damaged sample No. 5, the high value of dynamic modulus $E_D$ is observed ($E_D$ = 28.2 GPa). In contrast, in sample No. 9 with a large number of cracks, a low $E_D$ is detected ($E_D$ = 12.7 GPa). Based on these results, the five samples with different damage degrees are selected for future investigation by AE and DICM tests.

3. The fracture process behavior of concrete under compression is monitored by the AE technique. According to the results, all samples show different AE energy release trends which are affected by the uniqueness and severity of the internal defect system. Sample No. 5 with low crack density has a step-up tendency similar to non-damaged concrete, where with stress increasing, the AE energy release events appear and accumulate until final failure where sudden emission of high intensity occurs. In contrast, the AE energy release trend observed in sample No. 9 which has a complex crackling system demonstrates the continuous emission process starting from the early loading stage, and the cumulative AE curve has a steeper slope corresponding high rate of fracture. This result confirms that the damaged condition of the concrete structure can be detected by the AE measurement results.

4. Circumferential strain and radial displacement analyzed by the DICM show the correlation between strain localization points and the location of pre-existing defects. Because the defect is a weak point of the material structure, thus, it leads to the fracture processes evolving near this area. In samples with a high degree of cracking damage, the strain allocation occurs in multiple locations at the same time and causes an unstable deformation process.

5. Intensity of the AE signal in terms of amplitude and peak frequency bands and its spatial distribution within the concrete matrix was received by the AE source location analysis. Calculated values demonstrate that the high-amplitude low-frequency AE events are prevalent in all samples corresponding to the different fracture types. Location analysis shows the correlation between AE event occurrence and area of pre-existing defects: AE event localization density is higher in the defect location and matches its main pattern.

6. The image-gridding method was employed to deeply analyze the relationship between crack density and AE event characteristics. In areas with high relative crack density, the amplitude and peak frequency parameters of AE events have bigger values. In the adjacent with pre-existing cracks areas, more intensive AE signals are detected, and the crack development rate is higher regarding the potential crack propagation destinations. Moreover, the peak frequency is higher in samples with a considerable number of vertical cracks compared to samples with horizontal cracks.

7. The time-series analysis of AE and DICM showed the correlation between strain concentration and AE energy release. The strain allocation evidences the significant release of AE energy relating to fracture development and contributes to the main deformation processes. Increasing the deformation rate can be noticed after a significant AE energy release event.

It can be concluded that pre-existing defects resulting from the severe environmental condition and operation period affect the fracture behavior in the concrete under compression. The more crack density within the concrete matrix, the higher intensity of the AE events and strain concentration in the weak regions which can lead to future material deterioration and failure. Using a combination of AE and DICM techniques can give a comprehensive damage evaluation of the structure. These findings can be useful in monitoring the concrete structure condition and in identifying potential areas of defects that require attention to prevent failure or damage.

**Author Contributions:** Conceptualization, N.M.; methodology, N.M. and K.S.; validation, Y.S. and T.S.; data curation, K.S.; writing—original draft preparation, N.M.; writing—review and editing, Y.S. and T.S.; supervision, T.S. All authors have read and agreed to the published version of the manuscript.

**Funding:** This research received no external funding.

**Data Availability Statement:** The datasets collected and/or analyzed during the current study are available from the corresponding author on reasonable request.

**Conflicts of Interest:** The authors declare no conflict of interest.

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
