# Peer review of "Influence of the Pre-Existing Defects on the Strain Distribution in Concrete Compression Stress Field by the AE and DICM Techniques"

_applsci, doi:10.3390/app13116727_

Round 1
Reviewer 1 Report
In my opinion, it is an interesting article due to the fact that non-destructive tests for concrete are carried out. These types of tests are very necessary when carrying out infrastructure maintenance. The introduction and the methodology are adequate and the conclusions are clear and very understandable.Author Response
Dear Reviewer,
I am Nadezhda Morozova, the corresponding author of the article titled "Influence of the Pre-existing Defects on the Strain Distribution in Concrete Compression Stress Field by the AE and DICM techniques".
Thank you very much for your help and support of this research.
I believe that this article will contribute to the monitoring and analysis of in-service concrete structures and help engineers to choose an optimal maintenance way for them.
Best Regards,
Nadezhda Morozova
Reviewer 2 Report
In this manuscript, the authors presented the evaluation of the effects of the pre-existing defects on the strain distribution in concrete compression stress field by AE and DICM techniques, the concerns raised by the reviewer are shown as follows.
1. The characteristics of the pre-existing defects used for the research may need to be defined, such as the size, density, and distribution.
2. More detail discussion should be provided on how can the DICM results correlate with the AE results.
3. More information should be provided in the experimental part on how to obtain the time-series data from the DICM results.
4. The influence of the pre-existing defects on the strain distribution is not clearly shown in the context, the authors may need to either summarize the effects of defects on strain distribution and crack development in the conclusion part or change the word “influence” to “evaluation” in the title.
The language of the manuscript may need to be polished further, the frequency of the words such as “most”, “maximum” should be reduced.
Author Response
Dear Reviewer,
I am Nadezhda Morozova, the corresponding author of the article titled "Influence of the Pre-existing Defects on the Strain Distribution in Concrete Compression Stress Field by the AE and DICM techniques".
I appreciate the time and effort that you have dedicated to providing your valuable feedback on my manuscript.
I have been able to incorporate changes to the manuscript in accordance with your comments. Please see the attachment.
Best Regards,
Nadezhda Morozova

Reviewer 3 Report
This paper investigated the influence of preexisting defects within concrete taken from the inservice irrigation structure. Then, the X-ray Computed Tomography (CT) technique is employed to investigate the internal concrete matrix and evaluate the defect distribution. The cracking system in a concrete matrix is detected as a damage type caused by severe environment. The geometric properties of defects and the spatial location are obtained by image processing of CT images. The Digital Image Correlation technology and Acoustic Emission were used to obtain the displacement and strain fields and AE signals. Many useful conclusions are obtained. I can suggest its acceptance. Detailed comments are as follows:
1. Experimental scheme: It is recommended to list the test steps in accordance with the actual operation steps.
2. Speckle sample preparation, the speckle is crucial to the DIC result. However, the authors did not elaborate on the process of speckle making. In addition, there are strict standards for the density and manufacturing process of speckle. The paper can be referred to in this literature https://doi.org/10.1016/j.conbuildmat.2022.128838.
3. The distribution of the AE sensors and parameter settings are not introduced. The test results of AE are highly dependent on the sensor position and the parameter setting of the instrument. Please specify AE probe model, front gain and other related information.
4. Line 183: The Dynamic modulus of elasticity ED calculated from resonant frequency can be calculated by following Equation 1. Give the specific citations.
5. Figure 8. Circumferential strain exx and radial displacement dR. Under different working conditions, the reference amount of strain variety should be the same, rather than each being different. Otherwise, there is no significance in the comparison. For example, 3.4. Strain localization.
Author Response

(The authors gave the same response as above.)

Round 2
Reviewer 2 Report
In the abstract, it is more appropriate to mention the capacity of AE and DICM in evaluating the structural weakness in the concrete rather than the severity of fracture.
The english is fine but spell check is suggested before publication.
Author Response
Dear Reviewer,
I am Morozova Nadezhda, a corresponding author of this article.
We really appreciate your assistance and support of this research.
Thank you for pointing this aspect out. We agree with this and have incorporated your suggestion througthout the abstract (lines 13-26, pp.1)
Best Regards,
Nadezhda Morozova